# Aspartyl Protease Inhibitors as Anti-Filarial Drugs

**DOI:** 10.3390/pathogens11060707

**Published:** 2022-06-18

**Authors:** Liana Beld, Hyeim Jung, Christina A. Bulman, Bruce A. Rosa, Peter U. Fischer, James W. Janetka, Sara Lustigman, Judy A. Sakanari, Makedonka Mitreva

**Affiliations:** 1Department of Pharmaceutical Chemistry, University of California, San Francisco, CA 94158, USA; Liana.Beld@ucsf.edu (L.B.); cabulman@dons.usfca.edu (C.A.B.); 2Division of Infectious Diseases, Department of Medicine, Washington University School of Medicine, St. Louis, MO 63110, USA; jungh@wustl.edu (H.J.); barosa@wustl.edu (B.A.R.); pufischer@wustl.edu (P.U.F.); 3Department of Biochemistry and Molecular Biophysics, Washington University School of Medicine, St. Louis, MO 63110, USA; janetkaj@wustl.edu; 4Molecular Parasitology, New York Blood Center, Lindsley F. Kimball Research Institute, New York, NY 10065, USA; SLustigman@nybc.org; 5McDonnell Genome Institute, Washington University School of Medicine, St. Louis, MO 63108, USA

**Keywords:** Neglected tropical diseases (NTDs), filarial nematodes, macrofilaricidal, anti-filarial drugs, gastro-intestinal nematodes, *Trichuris*

## Abstract

The current treatments for lymphatic filariasis and onchocerciasis do not effectively kill the adult parasitic nematodes, allowing these chronic and debilitating diseases to persist in millions of people. Thus, the discovery of new drugs with macrofilaricidal potential to treat these filarial diseases is critical. To facilitate this need, we first investigated the effects of three aspartyl protease inhibitors (APIs) that are FDA-approved as HIV antiretroviral drugs on the adult filarial nematode, *Brugia malayi* and the endosymbiotic bacteria, *Wolbachia*. From the three hits, nelfinavir had the best potency with an IC_50_ value of 7.78 µM, followed by ritonavir and lopinavir with IC_50_ values of 14.3 µM and 16.9 µM, respectively. The three APIs have a direct effect on killing adult *B. malayi* after 6 days of exposure in vitro and did not affect the *Wolbachia* titers. Sequence conservation and stage-specific gene expression analysis identified Bm8660 as the most likely primary aspartic protease target for these drug(s). Immunolocalization using antibodies raised against the Bm8660 ortholog of *Onchocerca volvulus* showed it is strongly expressed in female *B. malayi*, especially in metabolically active tissues such as lateral and dorsal/ventral chords, hypodermis, and uterus tissue. Global transcriptional response analysis using adult female *B. pahangi* treated with APIs identified four additional aspartic proteases differentially regulated by the three effective drugs, as well as significant enrichment of various pathways including ubiquitin mediated proteolysis, protein kinases, and MAPK/AMPK/FoxO signaling. In vitro testing against the adult gastro-intestinal nematode *Trichuris muris* suggested broad-spectrum potential for these APIs. This study suggests that APIs may serve as new leads to be further explored for drug discovery to treat parasitic nematode infections.

## 1. Introduction

Neglected tropical diseases (NTDs) impact more than one billion people worldwide, primarily in resource-limited countries in tropical and sub-tropical regions [1]. Lymphatic filariasis (LF) is an NTD that is targeted for elimination; it is caused by the filarial nematodes *Brugia malayi*, *Brugia timori*, and *Wuchereria bancrofti* [2]. Chronic infection results in swollen legs, arms, and genitalia, which can lead to severe disability, social stigma, and decreased productivity [2]. Mass drug administration (MDA) for LF is administered to an entire at-risk population using a combination of two [3], or more recently, all three (“triple drug treatment”) of the mostly embryostatic or microfilaricidal drugs: albendazole, diethylcarbamazine citrate (DEC), and ivermectin [2,4,5]. These anthelmintics kill microfilariae over the lifetime of the adult worms (6–8 years for *Wuchereria* and *Brugia* spp. [6,7]). In 2018, an estimated 51 million people were infected with LF, which was about half the number of infected individuals in 2000 [8]. Although the global disease burden has declined due to MDA, LF elimination is still challenging in many areas and infection continues to lead to devastating and disabling illnesses in many individuals [9].

Elimination of LF has been challenging, due to the complex life cycle of the filarial worms and the inability of current treatments to clear the adult life stages of the filarial worms that cause the disease symptoms [8,10]. The World Health Organization recommends MDA for LF once per year for at least 5 years to cover the reproductive life span of the adult worms. Since compliance with MDA is suboptimal in many areas, many more rounds of MDA are often necessary to interrupt the transmission of microfilariae [4,5]. The presence of other co-endemic NTDs, specifically loiasis and onchocerciasis, determines which combination of drugs can be used [4]. Additionally, since the *Brugia* species rely on the endosymbiotic *Wolbachia* for development, fertility, and survival, *Wolbachia* is considered an excellent target for drug therapies [11,12]. Doxycycline is the primary treatment used to target *Wolbachia*, and although it is relatively inexpensive and readily available, patients must take it daily for 4 to 6 weeks, which prevents it from being easily administered through MDA [5]. Given that the recommended treatments do not effectively kill the adult filarial worms and there are complications based on the co-endemicity of other NTDs, the discovery of new and more efficacious drugs is critical.

One approach to identify new candidates to treat filarial diseases involves the use of repurposed drugs to screen for potential candidates [13]. The advantages of screening repurposed drugs are that many are already FDA-approved and considerable information is readily available on their safety and efficacy in humans, thus circumventing costly preclinical assays and making them more cost-effective [14,15,16]. Commercially available aspartyl protease inhibitors (APIs) have been shown to have macrofilaricidal effects in in vitro studies with adult *B. pahangi*, adult *O. ochengi* and L4 stages of *O. volvulus* [13], including APIs that are FDA-approved as antiretroviral drugs for human immunodeficiency virus (HIV) and acquired immunodeficiency syndrome (AIDS) such as lopinavir, nelfinavir, and ritonavir [17]. The target of the HIV antiretroviral drugs is the HIV-1 aspartyl protease, which is essential for the HIV to mature [17]. Aspartyl proteases belong to a class of proteolytic enzymes and are important in maintaining cellular functions, including cell fertilization, cell-to-cell interactions, and DNA replication and transcription [18]. Proteases dominate the list of targets for human diseases [19]; thus, conservation and diversification across the protease superfamily among parasitic nematodes has been explored [20]. For example, the expansion of protease families has been reported to occur due to differences in host tissue penetration (skin vs. oral) or feeding (digestion of blood vs. cellular content), but some level of conservation has also been detected [20]. Because polyparasitism (LF and onchocerciasis, or filarial and gastro-intestinal nematodes) is still widespread across rural communities in endemic countries [21], exploring the broad-spectrum potential of APIs is of great interest.

The purpose of this study is to expand the assessment of the efficacy of several APIs on the *Brugia* filarial nematode, to determine their effect on the *Wolbachia* endosymbiont, to immunolocalize the putative aspartic protease target in the adult *Brugia* worms, and to better understand worm responses to API treatment at the transcriptional level. The new findings confirmed that APIs warrant further study as new candidate leads for the filarial and gastro-intestinal nematode drug discovery program.

## 2. Materials and Methods

### 2.1. In Vitro Worm Motility Assays

#### 2.1.1. Adult Female B. malayi

Adult female *B. malayi* were obtained from the Filariasis Research Reagent Resource Center (FR3) at the University of Georgia, Athens. Each well in a 24-well plate contained one adult female *B. malayi* and media (RPMI-1640 with 25 mM HEPES, 2.0 g/L NaHCO_3_, 5% heat inactivated FBS, and 1X antibiotic/antimycotic solution) [13,22,23]. 10 mg of lopinavir (catalog #: S1380), nelfinavir (catalog #: S4282), and ritonavir (catalog #: S1185) were purchased from Selleckchem. A 10 mM stock solution was prepared with dimethyl sulfoxide (DMSO, Sigma-Aldrich D2650, St. Louis, MO, USA) and was diluted in culture media to achieve 1% DMSO in the final mixture. The negative control was 1% DMSO. The motility of the worms was measured using the Worminator instrument on days 0, 1, 2, 3, and 6 of exposure [23]. Each day, the plates of worms were recorded for 60 s using the Worminator, and the WormAssay application determined the number of pixels displaced per second for individual worms using the Lucas-Kanade optical flow algorithm (velocity, 1 organism per well) and calculated the mean movement units (MMUs) for each worm [23]. The percentage of inhibition of motility was calculated by dividing the MMUs of treated worms by the average MMUs of the control worms, then subtracting that value from 1 [23]. The values were floored to zero and multiplied by 100% [23].

Seven APIs were tested at single concentrations to determine the best hits (>75% inhibition of worm motility). Lopinavir and nelfinavir were tested at 30 µM and 10 µM. Amprenavir, atazanavir, and ritonavir were tested at 50 µM and 30 µM. Darunavir and pepstatin A were tested at 50 µM. The negative control was 1% DMSO. The worms were incubated at 37 °C with 5% CO_2_ for the duration of the assay [13].

Based on the worm assays that tested single concentrations over 6 days, the best hits were selected. IC_50_ assays were conducted for lopinavir, nelfinavir, and ritonavir using six-point dilutions (100 µM to 0.3 µM, 30 µM to 0.1 µM, and 300 µM to 1 µM, respectively). The negative control for this assay was 1% DMSO. The worms were incubated at 37 °C with 5% CO_2_ throughout the assay. Prism 6.0 was used to calculate the IC_50_ values with a non-linear regression curve [22,23].

#### 2.1.2. Adult T. muris

In vitro screening was performed as previously described [24], and adult *T. muris* were provided by Dr. Joseph Urban (USDA). On the day of arrival (day 0), adult worms were washed as described above and 2 worms were plated into each of 24-well plates containing 500 µL culture media. For each API treatment, 4 biological replicates were prepared. The APIs were tested at 50 µM and 100 µM, and control worms were treated with 1% DMSO and maintained at 37 ˚C with 5% CO_2_. The motility of the worms was observed daily on days 0 to 3 using the Worminator approach described above and the MMUs were estimated using the WormAssay software with the consensus voting luminance difference algorithm [23]. Prism 6.0 was used to calculate the IC_50_ values with a non-linear regression curve [22,23].

### 2.2. Determining Wolbachia Titers

To determine *Wolbachia* titers, adult female *B. malayi* were collected on day 1 when the worms were moribund after exposure to 100 µM lopinavir, 30 µM nelfinavir, and 30 µM ritonavir, respectively. Genomic DNA (gDNA) was extracted from *B. malayi* using the QIAGEN DNeasy Blood & Tissue Kit and the purity of the gDNA was quantified based on the 260/280 ratio (>1.8) using the NanoDrop™ One^c^ from Thermo Fisher Scientific.

The *Wolbachia* titers were determined by measuring the abundance of the single copy gene *Wolbachia* surface protein (*wsp*) using a quantitative real-time polymerase chain reaction (qPCR) [25,26]. The glutathione-S-transferase (*gst*) gene, a single-copy gene in the *Brugia* species, was used to normalize the *wsp*/*gst* ratio. The qPCR forward and reverse primers for *wsp* were 5′-CCCTGCAAAGGCACAA GTTATTG-3′ and 5′-CGAGCTCCAGCAAAGAGTTTAATTT-3′, respectively [25,26]. The qPCR forward and reverse primers for *gst* were 5′-GAGACACCTTGCTCGCAAAC-3′ and 5′-ATCACGGACGCCTTCACAG-3′, respectively [25,26].

The qPCR was performed using the Bio-Rad CFX Connect thermocycler with the following conditions for *wsp*: 95 °C for 15 min, 40 cycles of denaturation at 94 °C for 10 s, annealing at 57 °C for 20 s, and elongation at 72 °C for 15 s [25,26]. The melting curve analysis involved heating to 95 °C for 1 min, annealing at 55 °C for 30 s, then heating to 97 °C. The thermocycler conditions for *gst* were 95 °C for 15 min, 40 cycles of denaturation at 94 °C for 10 s, annealing at 57 °C for 20 s, and elongation at 72 °C for 15 s [25,26]. The melting curve analysis involved heating to 95 °C for 1 min, annealing at 55 °C for 30 s, then heating to 97 °C.

The standards were created by diluting 2 ng/µL of *wsp* plasmid and 2 ng/µL of *gst* plasmid to obtain known copy numbers of the *wsp* and *gst* plasmid. The copy numbers of the standards were plotted against the Ct values from the thermocycler results using MS Excel, and a line of best fit was calculated. Prism 6.0 was used to analyze the data with a Kruskal–Wallis non-parametric ANOVA test.

### 2.3. Primary Sequence-Based Comparisons of Aspartyl Proteases

Protein sequence data was downloaded from WormBase Parasite [27]: *B. malayi* (PRJNA10729), *B. pahangi* (PRJEB497), *O. ochengi* (PRJEB1204), *O. volvulus* (PRJEB513), *T. muris* (PRJEB126), and *Caenorhabditis elegans* (WBPS16). In addition, the human genome (GRCh38.p13), downloaded from Ensembl [28], was used as an outgroup for comparisons. Aspartyl proteases of *Brugia* spp., *Onchocerca* spp. and *T. muris* were annotated using MEROPS [29]. All of the *B. malayi* aspartyl proteases identified by MEROPS were aligned with HIV-1 protease (GenBank accession number: CAA09224) using T-Coffee [30] and clustered with Clustal Omega [31]. Orthology was determined among the 6 nematode species with human using Orthofinder with default settings [32]. Orthologous protein families annotated as aspartyl proteases in *Brugia spp.*, *Onchocerca spp.*, and *T. muris* were selected for multiple sequence alignment; construction of a phylogenetic tree using the neighbor joining method was performed with MAFFT version 7 with default setting [33]. The alignment was visualized with Jalview [34].

Best bi-directional hits were identified based on protein alignment between the longest isoforms per gene of *B. malayi* and those of *B. pahangi* using Diamond blastp (v2.0.6.144) [35]. This alignment was performed in “both directions” (once with *B. malayi* as the query and once with *B. pahangi* as the query) and a local script was used to build a table of results, with all cases in which proteins were top scoring hits next to each other in both directions and flagged as being reciprocal hits. Gene symbols and descriptions were harvested from the BioMart [36] instance hosted at WormBase Parasite [27] and were used to annotate the results.

### 2.4. Histochemical Localization of the Aspartyl Proteases in Adult Female B. malayi 

#### 2.4.1. Immunohistology

A polyclonal rabbit antibody raised against a recombinant cathepsin D-like lysosomal aspartic protease of the filarial parasite *O. volvulus* (GenBank accession number U81605) was used as a primary antibody to label an orthologous protein in *B. malayi* [37]. Twelve-week-old adult *B. malayi* recovered from experimentally infected gerbils were fixed in 80% ethanol and stained using the alkaline phosphatase/anti-alkaline phosphatase technique as described previously [38].

#### 2.4.2. Immunogold Transmission Electron Microscopy

For immunolabeling, we used high-pressure frozen *B. malayi* samples as described previously [39]. The primary anti-polyclonal rabbit antibody was used at dilutions of 1:50 and 1:200 and as negative control, phosphate buffered saline was used instead of the primary antibody. To visualize the primary rabbit antibody, an 18-nm colloidal gold-conjugated goat anti-rabbit secondary antibody (Jackson ImmunoResearch, West Grove, PA, USA) was used and sections were viewed on a JEM-1400 transmission microscope (JEOL) at 80 kV with an AMT XR111 4 k digital camera.

### 2.5. RNAseq of Adult Female B. pahangi In Vitro Treated with APIs and Comparative Transcriptomics

#### 2.5.1. Prepartion of In Vitro Treated Worms with APIs and Production of RNA-seq

For RNA-seq analysis, adult female *B. pahangi* (3 worms per replicate) were treated with 1% DMSO (control) and 100 µM of lopinavir, nelfinavir, and ritonavir for 1 h of in vitro treatment (all in triplicate) and stored at −80˚C. Total RNA was isolated using Trizol (Invitrogen, Waltham, MA, USA) after being homogenized with 1.5 mm zirconium beads (Benchmark scientific, D1032-15) using BeadBug6 (Benchmark scientific, D1036, Sayreville, NJ, USA). cDNA libraries were prepared from RNA samples with poly(A) enrichment and processed cDNA was sequenced on the Illumina NovaSeq S4 platform (paired-end 150bp reads). Sequenced reads were mapped to the *B. pahangi* genome assembly (PRJEB497, WBPS13 [27]) using STAR [40] and quantified per gene using featureCounts [41]. Relative gene expression was calculated using “fragments per kilobase per million reads” (FPKM) normalization per gene per sample. Fold change and differential expression significance values (relative to DMSO controls) were calculated using DESeq2 [42] (version 1.34.0; default settings, FDR ≤ 0.05). DESeq2 [42] was also used to perform PCA (default settings, based on the top 500 most variable genes). The raw RNA-seq read files (fastq) are accessible on the NCBI Sequence Read Archive (SRA [43], BioProject PRJNA840851). Complete sample metadata, RNA-seq mapping statistics, and accession numbers per sample are provided in Appendix A. Read counts and normalized expression values per gene per sample are provided in Appendix A.

Gene expression profiles based on stage-specific RNA-seq datasets were generated using the same processing approach applied to datasets from Choi et al. [44] for *B. malayi*, and Foth et al. [45] for *T. muris*.

#### 2.5.2. Functional Annotation of B. pahangi Genome for RNA-seq Analysis

Functional annotations for all *B. pahangi* genes were assigned using annotations from InterProScan v5.42 [46] to identify gene ontology [47] classifications and InterPro functional domains [48], and GhostKOALA v2.2 [49] to identify KEGG [50] annotations. MEROPS [29] was used to identify 14 aspartic proteases in the *B. pahangi* genome. All functional annotation data are provided in Appendix A. Enriched KEGG pathways among differentially expressed gene sets were identified using WebGestaltR [51] (minimum 2 genes per pathway, *p* ≤ 0.05 threshold for significance).

## 3. Results

A previous study reported some HIV protease inhibitors that have macrofilaricidal activity with stage and species specificity, identifying three aspartic protease inhibitors (APIs) that effectively inhibited motility in adult female *B. pahangi* and adult *O. ochengi* male and/or female worms [13]. To expand on these results, we first performed single-dose phenotypic screening on adult female *B. malayi* (human parasite) followed by determining IC_50_ values, evaluating whether the efficacy is via the *Wolbachia* endosymbiont or through direct effects on the worm, performing immunolocalization of the potential aspartic protease target, and interrogating the adult worm transcriptional response to treatment.

### 3.1. In Vitro Worm Motility Assays

Guided by the Tyagi et al. study [13], seven FDA-approved APIs were screened against adult female *B. malayi* at either 30 µM or 50 µM in a motility assay (Figure 1A, Appendix A). The worms treated with lopinavir, nelfinavir, and ritonavir showed the highest motility inhibition at day one and consequently they were killed by day 6 (Figure 1A). Worm motility was not inhibited by four (amprenavir, atazanavir, darunavir, and pepstatin A) of the seven APIs, as previously shown in adult *B. pahangi* and *O. ochengi* [13]. However, the three effective APIs had similar efficacy at 30 µM in both our *B. malayi* assay and the previous *B. pahangi* assay after 6 days [13] (Figure 1B). To further investigate the potency of the drugs, the IC_50_ values for lopinavir, nelfinavir, and ritonavir were determined to be 16.94 µM, 7.78 µM, and 14.30 µM, respectively, on day 6 in adult female *B. malayi* (Figure 1C–E). Based on the estimated IC_50_ values, nelfinavir was the most potent drug against adult female *B. malayi* in vitro. The IC_50_s were better than or in the same range as those reported for *O. ochengi* female (nelfinavir and ritonavir, 19 and 16 µM, respectively; IC_50_ on lopinavir against female *O. ochengi* has not been determined [13]).

### 3.2. Determining Wolbachia Titers

*Wolbachia* titers were analyzed to determine if the APIs also affected the *Wolbachia* population in the adult female *B. malayi*, since *Wolbachia* are necessary for filarial worm survival [52]. No significant difference was observed in the copy number ratio of *Wolbachia* surface protein (*wsp*)/*Brugia* glutathione-S-transferase gene (*gst*) in the worms treated with each of 100 µM lopinavir, 30 µM nelfinavir, and 30 µM ritonavir, compared with the DMSO controls (Figure 2; differences tested using a two-tailed T-test with unequal variance). This result suggests that the anti-filarial activity of the three APIs is directly affecting the adult worms rather than the endosymbiotic bacteria, *Wolbachia*.

### 3.3. Primary Sequence Comparison of Aspartyl Proteases in Filarial Worms

Using MEROPS annotation, we identified 16 APs in *B. malayi* and found that three genes (Bm8660, Bm7467, and Bm3492) are the most homologous to the HIV-1 AP (Figure 3A). Bm8660, encoding lysosomal cathepsin D, was one of the genes over-expressed in later stages of the life cycles, including adults, and conserved across different species as single-copy orthologs: *B. pahangi* (BPAG_0001244901), *O. volvulus* (OVOC11635), *O. ochengi* (OOCN_959601), *T. muris* (TMUE_3000010755), human (CTSD), and *C. elegans* (asp-4) (Figure 3B, Appendix A). Multiple sequence alignment showed evolutionary conservation of the orthologous lysosomal proteases and identified two aspartic protease catalytic triads (DTG), compared with single motif in the HIV-1 AP (Figure 3B). In HIV and other species, the aspartic protease active site is reconstituted by homodimerization of the protein and requires critical DTG or DSG motifs, one from each monomer. As previously noted, the entire active site of the aspartic proteases in nematodes appears to be located in the same protein and does not require dimerization for activity, unlike HIV [13].

### 3.4. Immunolocalization of Lysosomal Aspartic Protease in Adult Female B. malayi

Immunolocalization using a polyclonal Ov-APR (OVOC11635, ortholog of Bm8660) antibody shows that this aspartic protease is strongly expressed in the lateral and dorsal/ventral chords, the hypodermis, intrauterine stretched microfilariae, and the uterine wall of adult female *B. malayi* (Figure 4A). Intense labeling was also observed in the intestinal wall, but unlike all other structures, staining was also seen in the negative control that lacked the primary antibody, because of intrinsic alkaline phosphatase in the intestine. Immunogold transmission electron microscopy showed a distinct staining pattern (Figure 4B–F). For example, no labeling was observed in the cuticle, the muscles, mitochondria, or *Wolbachia* endobacteria, while extensive labeling of vacuoles within the hypodermis was seen (Figure 4C,F). The density of lysosomes in filarial parasites increases with age, and we were not able to detect characteristic, electron-dense, non-depleted lysosomes in the young adult females examined. However, we could sometimes observe large clusters of gold particles associated with moderately electron-dense structures within the cytoplasm of the lateral chords (Figure 4E). These immunolocalization experiments show that Bm8660 is strongly expressed in female *B. malayi*, especially in metabolically active tissues such as lateral and dorsal/ventral chords, hypodermis, and uterus tissue.

### 3.5. Transcriptional Response of API-Treated Adult Female B. pahangi

The transcriptional response to API treatment was quantified using RNA-seq analysis of adult female *B. pahangi* worms treated with 1% DMSO (control) and 100 µM of lopinavir, nelfinavir, and ritonavir for 1 h (in vitro treatment, performed in triplicate). Sequencing produced an average of 36 million reads, with an average of 92.6% of reads mapping to the *B. pahangi* genome, following cleaning and processing (Appendix A). The exposure time and concentration were selected based on the determined IC_50_ values in order to optimally capture the immediate transcriptional response, including that of aspartic protease-responsive genes, before downstream stress-response pathways were activated.

Principal components analysis (PCA) of RNA-seq datasets showed separate clustering of the API-treated worms from the untreated group, suggesting the early transcriptomic responses from the API-treated worms were distinct from those of the controls (Figure 5A). Differential expression analysis identified significantly upregulated/downregulated genes in lopinavir (upregulated: 204, downregulated: 105), nelfinavir (upregulated: 869, downregulated: 760), and ritonavir (upregulated:1226, downregulated: 1257) compared to DMSO controls, respectively. Among them, more than 40% of differentially expressed genes were shared with at least two comparison groups (upregulated: 49.2%, downregulated: 40.2%), particularly between nelfinavir and ritonavir. This indicated that API treatments could elicit common and unique transcriptional responses in adult female *B. pahangi* (Figure 5B). Four of the 14 APs identified by MEROPS were differentially regulated by the APIs (Figure 5C). Of these, two genes encoding cathepsin E (BPAG_1278201) and signal peptide peptidase-like protein (BPAG_106301) were significantly upregulated by ritonavir. A presenilin 1 gene (BPAG_1346801)—a critical regulator of ErbB/EGFR [53] signaling pathway—was downregulated by both ritonavir and nelfinavir. Interestingly, a DNA damage-inducible 1 homolog (BPAG_1342901) whose activity can be directly inhibited with nelfinavir-treated cancer cells [54,55], was significantly downregulated only in nelfinavir. Gene annotations, read counts, normalized expression values, and differential expression statistics for all genes in all samples are provided in Appendix A. 

Next, we identified significantly enriched KEGG pathways among upregulated/downregulated genes in the API treatments (Figure 6). Most enriched pathways were observed in nelfinavir and ritonavir, due to the lack of differentially expressed genes in lopinavir. We found that KEGG pathways involved in ubiquitin mediated proteolysis, protein kinases and MAPK/AMPK/FoxO signaling pathways were significantly enriched among upregulated genes in all API treatments (Figure 6A). Upregulation of genes related to the ubiquitin mediated proteolysis may be related to potential inhibition of the filarial lysosomal AP (Figure 3C,E,H) by APIs, consequently affecting protein degradation and increasing the burden of protein aggregates. The ubiquitin proteasome system and autophagy-lysosomal pathway constitute a single network to regulate proteostasis [56]. In addition, lopinavir, nelfinavir and/or ritonavir have been shown to trigger an accumulation of unfolded proteins, subsequently inducing an unfolded protein response (UPR) and ER stress, which can result in cell cycle arrest and/or apoptosis in mammalian cells [57,58,59,60,61,62,63]. Signaling pathways such as MAPK are evolutionarily conserved mechanisms for regulating cellular proliferation, differentiation, and survival, and can be activated via various extracellular stimuli [64]. We observed that genes involved in proteolysis and the several signaling pathways were immediately modulated in *B. pahangi* after the API treatments, which were corroborated by similar findings in previous mammalian cells [57,58,59,60,61,62,63].

### 3.6. In Vitro T. muris Adult Worm Motility Assays

While the overall repertoire of aspartic proteases in filarial worms and gastro-intestinal nematodes is different [20], there are orthologous proteins of the potential targets of the APIs in the whipworm *T. muris* (Figure 7A). This prompted us to test the activity of the three API hits in the *Brugia* species (Clade III) against the phylogenetically distant gastro-intestinal nematode *T. muris* (Clade I). Two of the APIs showed over 70% inhibition already at day 1, and all three inhibited the motility of adult whipworm by day 3 (Figure 7B), indicating that the conserved A01A aspartic proteases (especially TMUE_3000010755, which belongs to the same orthologous family [OG0004684] with Bm6880 and OVOC11635; Appendix A) might be a potential target in this species as well.

## 4. Discussion

Lymphatic filariasis (LF) is a neglected tropical disease affecting 51 million people worldwide [8]. Currently, there are no vaccines or drugs that can efficiently clear the adult worms that produce microfilariae necessary for transmission [65,66]. Aspartyl protease inhibitors that are FDA-approved as HIV antiretroviral drugs were investigated to potentially identify new drug therapies from repurposed compounds to treat LF.

The previous study implicated the repurposing of several APIs as potential macrofilaricides, based on the motility inhibition results in adult female *B. pahangi*, which are laboratory rodent filarial parasites [13]. Interestingly, lopinavir did not affect the viability of adult female *O. ochengi* (a cattle parasite closely related to the agent of human river blindness), but showed macrofilaricidal potency on male *O. ochengi* and female *B. pahangi*, suggesting a sex- and/or species-specific mechanism of action [13]. Here, we showed that the three prioritized APIs (lopinavir, nelfinavir, and ritonavir) effectively inhibited the motility of adult female *B. malayi* and consequently killed the adult worms in vitro (Figure 1A). This corroborated the potential translation of the findings from animal to human filarial worms. Although we did not confirm the motility inhibition of adult male and earlier stages of *B. malayi*, our findings support the potential usage of APIs as macrofilaricides to efficiently control lymphatic filariasis and/or their usage as lead compounds for further modifications to improve efficacy. These three hits are already FDA-approved compounds as HIV antiretroviral drugs, so product information on the pharmacokinetics and clinical data is readily available [17].

Determining *Wolbachia* titers in adult female worms provided insight on whether these endosymbiotic bacteria were affected by the APIs. Previous studies showed that because filarial worms including the *Brugia* species are dependent upon *Wolbachia* for survival and fecundity, eliminating the endosymbiotic bacterium would result in worm death [5,11,12,67]. However, the qPCR results showed that the APIs lopinavir, nelfinavir, and ritonavir had no effect on the *Wolbachia* titers when compared with the control, suggesting that the APIs are likely not affecting the *Wolbachia*, but instead have a direct effect on the worms.

To identify the potential targets of the APIs in filarial worms, we first conducted protein sequence-based clustering of the APs in *B. malayi* with HIV-1 aspartyl protease, followed by transcriptomic profiling across *B. malayi* life stages. We identified that some APs showed stage- and/or sex-specific expression, but two of the three APs (all belong to the A01A class of aspartic proteases) that were the most homologous to HIV1-PR (Bm8660: lysosomal aspartic protease or cathepsin D and Bm3492) were particularly over-expressed in L4 and adult stages, which potentially implicated their important roles in these clinically relevant stages (Figure 3A). This observation was also validated by immunolocalization of the lysosomal AP in adult *B. malayi*. This protein was specifically localized in large vacuoles of the hypodermis and the lateral and dorsal/ventral chords of adult female worms (Figure 4A–F). The density of typical, smaller lysosomes increases with the age of the worms and no lysosomes were found in the examined sections of the very young adult stage females used. Additionally, the lysosomal AP was identified in an orthologous protein family conserved in different species, including the filarial worms, whipworm (*T. muris*), *C. elegans*, and human (Figure 3B). Intriguingly, the orthologous lysosomal AP genes were identified as single-copy orthologs, while the orthologous genes to Bm3492 seemed to undergo several duplication events, especially in *C. elegans* and human (Appendix A). These results imply strict conservation of the lysosomal APs, given the last common ancestor and the indispensable functional role of the APs in protein homeostasis across different species.

The primary goal of the RNA-seq analysis was to capture the early transcriptional responses of adult female *B. pahangi* after the in vitro API treatments, in the hope that we could identify differential gene responses to the aspartic proteases and identify molecular pathways affected by treatment. The differential expression analysis showed that four of the 14 AP genes in *B. pahangi* were differentially expressed in nelfinavir and/or ritonavir, but not in lopinavir (Figure 5C). Lopinavir showed the least transcriptional responses, compared with nelfinavir and ritonavir, possibly because strong transcriptional changes may occur more gradually with this drug (Figure 5B). One of three biological replicates closely clustered with the untreated group (thus, considered as an outlier and removed from downstream analysis) and the other two replicates had a lower correlation coefficient compared with the replicates of the other two APIs, resulting in the under-detection of DEGs in lopinavir-treated worms. It is possible that in vitro treatment of lopinavir at 100 µM for 1 h may not be sufficient to trigger significant transcriptional changes in adult female *B. pahangi*; the motility of the worms after the treatments was not critically affected (data not shown).

Of the four AP genes differentially expressed in the API treatment, BPAG_1278201 (predicted as cathepsin E; CTSE) was found to be significantly upregulated by ritonavir (Figure 5C), which has been shown to have a high affinity for CTSE binding in previous research, which utilized it as a marker for CTSE detection [68,69]. BPAG_201201 (presenilin 1, PS1), predicted to be involved in regulating ErbB signaling receptor turnover, was downregulated by both ritonavir and nelfinavir (Figure 5C) [53]. Furthermore, our pathway enrichment analysis indicated upregulation of genes in the ErbB/EGFR pathways in both treatments with simultaneous downregulation of this PS1 gene (Figure 5C and Figure 6A). Finally, BPAG_1342901 (a DNA damage-inducible 1 homolog, DDI-1 or DDI-2) was significantly downregulated only in nelfinavir (Figure 5C). Previous studies demonstrated that DDI-2 protease resembled the 3D structure of the HIV protease, and nelfinavir could interfere with the proteasomal activity of DDI-2 [54,55,70]. Along with lysosomal protein degradation, the ubiquitin-proteasome system is a part of major proteolytic systems that play essential roles in degrading unnecessary proteins polyubiquitinated by ubiquitin [56]. In the immunohistological analysis, the highly expressed *B. malayi* lysosomal aspartic protease Bm8660 (which is the ortholog of the known API target in HIV) was identified in large vacuoles and other electron-dense structures in the cytoplasm, suggesting that is it potentially highly abundant in lysosomes, as expected based on results from other species. The RNA-seq results also show enrichment for common highly-expressed lysosomal-associated pathways among differentially expressed genes, further supporting this identification. However, since young adult worms were used, the exact positions of lysosomes could not be determined. The dysregulation of the protein homeostasis can result in activation of unfolded protein response (UPR) and ER stress in the cells [59]. Interestingly, we observed upregulation of genes involved in the ubiquitin-mediated proteolysis pathway and the various kinase signaling pathways (Figure 6), which may be cellular survival cues to resolve UPR and ER stress caused by the burden of accumulation of unfolded proteins.

In summary, this study demonstrated that the three APIs (lopinavir, nelfinavir, and ritonavir) originally designed to treat HIV infection could be repurposed to interrupt the transmission of filarial and/or gastro-intestinal infections by inhibiting the motility of adult worms. We also suggested a lysosomal aspartic protease as a potential target of the active API(s) based on the observations from protein sequence conservation, characteristic expression signatures at L4 and adult stages, immunolocalization, and drug-responsive transcriptional profiles from adult female worms. However, the specific molecular target of HIV protease inhibitors is still unclear; suggested potential targets include not only HIV-1 aspartic protease, but also heat-shock protein 90 (HSP90), cytochrome P450 3A4 (CYP3A4), and/or P-glycoprotein [71]. Therefore, in addition to the lysosomal aspartic proteases suggested in this study, the evaluation of other potential molecular targets should be considered in future experiments. Future studies are needed to confirm the molecular targets of these APIs and to help in understand the essentiality of these targets for parasitic nematode survival. While, historically, there has been a weak relationship between hits discovered in vitro and their efficacy in vivo [72], selective drug optimization could be followed by in vivo activity confirmation using animal models for filarial infection (jirds) and gastro-intestinal whipworm (mice) using human-equivalent dosages that are predicted based on animal PK data and the human safety and clinical information of the FDA-approved APIs.

## 5. Conclusions

Overall, this project has contributed to the discovery of potential new scaffolds that can be further explored to identify lead compounds that need to be further optimized. *B. malayi*, one of the parasitic nematode species that causes lymphatic filariasis and the gastro-intestinal whipworm, *T. muris*, have several aspartyl proteases that are conserved with the HIV-1 aspartyl protease and that are over-expressed in the adult female life stage. Thus, these aspartyl proteases may be the targets of the HIV antiretroviral drugs. This study revealed that the APIs lopinavir, nelfinavir, and ritonavir have a direct effect in killing the adult stages of two phylogenetically very different parasitic nematodes. This study also provided new insights on the filarial response to treatment at the transcriptional level. Future in vitro studies are necessary to validate the target of these APIs, to optimize the API structures to increase potency, and to follow up with in vivo studies for safety and pharmacokinetics, eventually leading to in vivo studies in animal model for filarial and whipworm infection. Furthermore, additional studies are necessary to determine the therapeutic indices and physiological pharmacokinetics for these drugs as anti-filarial drugs. Overall, the results of this study suggest APIs may serve as new leads for drug discovery of filarial, gastro-intestinal, and potentially other diseases.

## Figures and Tables

**Figure 1 pathogens-11-00707-f001:**
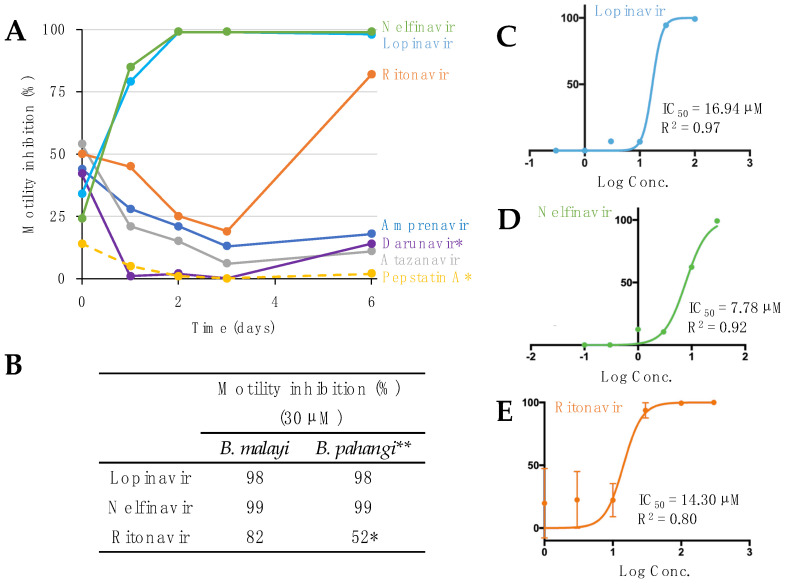
Aspartic protease inhibitor effects on motility in *B. malayi*. (**A**) Motility inhibition (% relative to DMSO) of adult female *B. malayi* treated in vitro with aspartic protease inhibitors over the course of 6 days. * All compounds treated at a concentration of 30 µM, except darunavir and pepstatin A, which were at concentrations of 50 µM. (**B**) After 6 days of treatment, the three APIs effective on *B. malayi* were also effective on *B. pahangi*. * Ritonavir was at 10 µM in *B. pahangi*; ** Results from Tyagi et al., 2021 [13]. IC_50_ values of (**C**) lopinavir, (**D**) nelfinavir, and (**E**) ritonavir with adult female *B. malayi,* based on in vitro treatment for 6 days at six different concentrations, and four replicates per concentration.

**Figure 2 pathogens-11-00707-f002:**
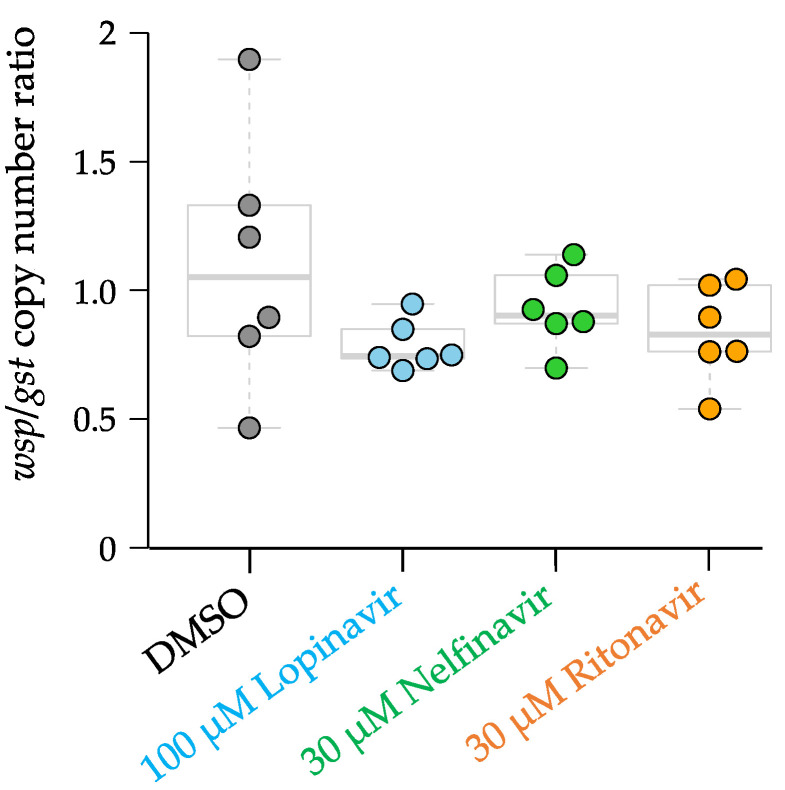
*Wsp/gst* ratio of adult female *Brugia malayi*. Adult female *B. malayi* were treated with 100 µM lopinavir, 30 µM nelfinavir, and 30 µM ritonavir and collected on day 1 of the assay.

**Figure 3 pathogens-11-00707-f003:**
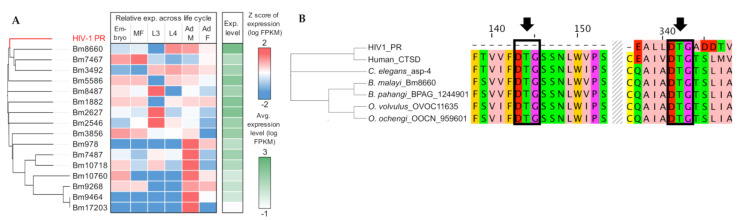
Sequence conservation and gene expression analysis of the *B. malayi* aspartic protease Bm8660. (**A**) Protein sequence-based clustering of the HIV-1 protease and all of the *B. malayi* aspartic proteases identified by MEROPS (sequence alignment using T-Coffee, clustering with Clustal Omega). The relative gene expression (Z score of log FPKM) and average absolute expression level (log FPKM, in green) of aspartic proteases across the life cycle are shown, based on RNA-seq data collected from Choi et al., 2011 [44]. Bm8660 was the closest *B. malayi* ortholog to HIV1 protease and had high expression in L4 and adult stages. (**B**) Sequence-based alignment and clustering of top-hit orthologs of Bm8660 among filarial worms, *C. elegans* (the lysosomal aspartic protease ASP-4), and human (Cathepsin D, CTSD). Arrows indicate two conserved aspartic protease catalytic triads (DTG).

**Figure 4 pathogens-11-00707-f004:**
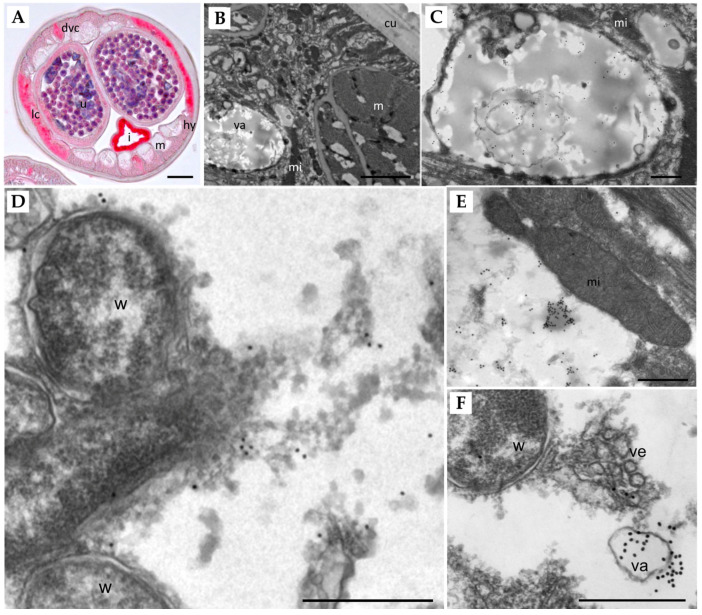
Localization of Bm8660 in adult female *B. malayi*. (**A**) Immunohistological stain using an Ov-APR antibody of an entire cross-section shows strong red staining in the lateral and dorsal/ventral chords, the hypodermis, intrauterine stretched microfilariae, the uterine wall, and the intestinal wall. ((**B**–**F**) Immunogold TEM labeling of APR. (**B**) Section of the hypodermis showing parts of the cuticle, muscles, mitochondria, and a large vacuole. (**C**) Magnification of (**B**) showing multiple clusters of gold particles within the large vacuole. (**D**) *Wolbachia* in the lateral chord are not labeled by gold particles, but small clusters of particles are found in nearby cytoplasm. (**E**) Mitochondria are also not stained by the gold particles, but a large cluster of particles can be seen in moderately electron-dense structures in the cytoplasm. (**F**) A large cluster of gold particles is associated with a vacuole in the vicinity of vesicles released by *Wolbachia* endobacteria. Lc, lateral chord; dvc, dorsal/ventral chord; u, uterus; I, intestine; m, muscles; hy, hypodermis; va, vacuole; mi, mitochondrion; cu, cuticle; w, Wolbachia; ve, vesicles. Scale bar: (**A**) 25 µm, (**B**) 2 µm, (**C**–**F**) 500 nm.

**Figure 5 pathogens-11-00707-f005:**
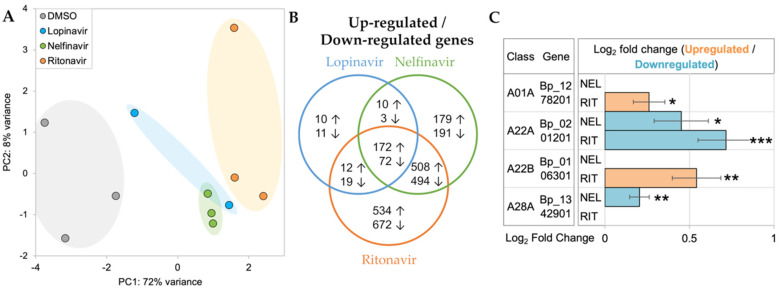
RNA-seq analysis of adult female *B. pahangi* treated with 1% DMSO, 100 µM of lopinavir, nelfinavir, and ritonavir in vitro for 1 h. (**A**) Principal components analysis (PCA) of RNA-seq samples based on gene expression patterns across all genes. (**B**) Differential expression analysis using DESeq2 identified significantly (FDR ≤ 0.05) upregulated (↑) and downregulated (↓) genes in API treatments relative to 1% DMSO controls. (**C**) Significantly differentially expressed APs. “Class” refers to the MEROPS class annotation. Log2 fold change values are relative to DMSO control. * *p* ≤ 0.05, ** *p* ≤ 0.01, *** *p* ≤ 0.001, after FDR adjustment, according to DESeq. NEL = nelfinavir; RIT = ritonavir; Bp_1278201 = BPAG_ 1278201; Bp_0201201 = BPAG_ 0201201; Bp_0106301 = BPAG_ 0106301; Bp_1342901 = BPAG_ 1342901.

**Figure 6 pathogens-11-00707-f006:**
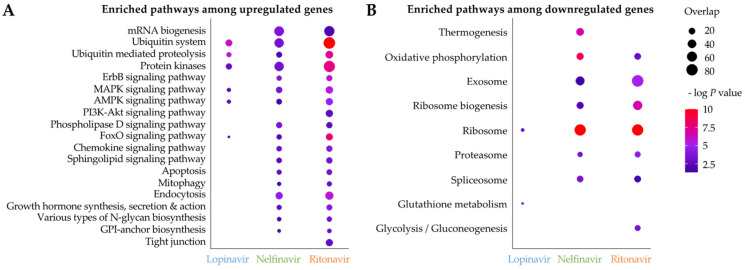
Significant KEGG pathway enrichment among (**A**) genes upregulated and (**B**) genes downregulated by API treatments in adult female *B. pahangi* (100 µM for 1 h). The –log *p* value (FDR-corrected) for each pathway and each gene is represented by the color, and the number of significantly differentially expressed genes from each pathway is represented by the dot size.

**Figure 7 pathogens-11-00707-f007:**
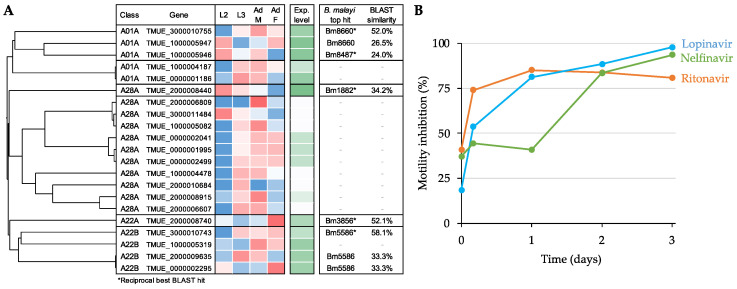
Aspartic protease analysis using the whipworm *Trichuris muris*. (**A**) Protein sequence-based clustering of the aspartic proteases identified by MEROPS (sequence alignment using T-Coffee, clustering with Clustal Omega). The plot excludes 27 A11A and 38 A28B class aspartic proteases. The relative gene expression (Z score of log FPKM) and the average absolute expression level (log FPKM, in green) of aspartic proteases across the life cycle are shown, based on RNA-seq data collected from Foth et al., 2014 [45]. *B. malayi* orthologs of *T. muris* APs with a BLAST E value < 10^−5^ are indicated, along with the amino acid sequence similarity (%) of the match. (**B**) Motility inhibition (% relative to DMSO) of adult *T. muris* treated in vitro with APIs (50 µM over the course of 3 days).

## Data Availability

All unprocessed RNA-seq data are available on the NCBI Short Reads Archive (SRA, BioProject PRJNA840851). Accessions per sample are available in Appendix A and complete-processed RNA-seq read counts, gene expression values, and annotations are available in Appendix A.

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
