# Peer review of "Aspartyl Protease Inhibitors as Anti-Filarial Drugs"

_pathogens, 2022, doi:10.3390/pathogens11060707_

Round 1

Reviewer 1 Report

This paper is well presented, with suitable use of the language and presented in a clear and objective manner. The authors bring some new insights into the potential use of some of the anti-Hiv APIs, deepening the results first presented by Tyagi et al. 2021. The results are relevant and interesting and the methods, until I'm able to opine about them, are suitably employed. 

With the aim of adding up to the quality of the paper, I suggest the following minor revisions:

1. Line 76: It is not possible to relate reference 15 to the paragraph in which it appears. The reference is about what exactly FDA approves or not and the paragraph is discussing the advantages of employing drug repurposing. See if a better reference could be used instead.

2. Line 81: again, reference 16 seems not to be related to what is discussed in the paragraph... 

3. Line 165: it would be interesting if the authors could describe a little bit more about the sequence comparison procedure, to allow other researchers to perform similar studies directly from their reference. 

4. Line 235: The supplementary table title is not correct. It is presented as "Supplementary Table S1: RNA-seq sample metadata including read mapping statistics and raw data accession identifiers." while the results are about % of inhibition of motility. Moreover, it would be important to explain what exactly the dash ("-") on the table is about. For instance, it is corresponding to "not tested" or "not active"?

5. Figure 1A. It would be interesting to investigate the ritonavir different behaviour since it seems not to affect the worm in the first 3 days, being effective only after day 4. Could it be related to a previous metabolism by the nematode organism or, any necessary transformation? THIS POINT IS A SUGGESTION FOR FUTURE STUDIES, NOT REQUIRED TO APPROVE THIS PAPER.

6. Finally, the authors suggest that the tested APIs would be suitable for drug repurposing, but I would be a little bit more careful since drug repurposing requires more than being active against a new target... It is also about being efficient and safe at dosages near those ones already used in the current therapy. For instance, in this paper, the authors are exploring anti-HIV drugs with known side effects with an impact on the patient's life quality. This same impact is acceptable for LF patients?

Author Response

With the aim of adding up to the quality of the paper, I suggest the following minor revisions:

  1. Line 76: It is not possible to relate reference 15 to the paragraph in which it appears. The reference is about what exactly FDA approves or not and the paragraph is discussing the advantages of employing drug repurposing. See if a better reference could be used instead.

Authors’ response: The reference has been replaced with more relevant ones:

Alavi SE, Ebrahimi Shahmabadi H. Anthelmintics for drug repurposing: Opportunities and challenges. Saudi Pharm J. 2021 May;29(5):434-445. doi: 10.1016/j.jsps.2021.04.004. Epub 2021 Apr 16. PMID: 34135669; PMCID: PMC8180459.

Panic G, Duthaler U, Speich B, Keiser J. Repurposing drugs for the treatment and control of helminth infections. Int J Parasitol Drugs Drug Resist. 2014 Jul 30;4(3):185-200. doi: 10.1016/j.ijpddr.2014.07.002. PMID: 25516827; PMCID: PMC4266803.

  1. Line 81: again, reference 16 seems not to be related to what is discussed in the paragraph... 

Authors’ response: Reference 16 has been removed, the relevant reference is #17 and has been retained.

  1. Line 165: it would be interesting if the authors could describe a little bit more about the sequence comparison procedure, to allow other researchers to perform similar studies directly from their reference. 

Authors’ response: more details have been provided in section 2.3, as follows:

2.3. Primary Sequence-Based Comparisons of Aspartyl Proteases

Protein sequence data was downloaded from WormBase Parasite [28]: B. malayi (PRJNA10729), B. pahangi (PRJEB497), O. ochengi (PRJEB1204), O. volvulus (PRJEB513), T. muris (PRJEB126), and Caenorhabditis elegans (WBPS16). In addition, the human genome (GRCh38.p13) was used as an outgroup for comparisons and was downloaded from Ensembl [29]. Aspartyl proteases of Brugia spp., Onchocerca spp. and T. muris were annotated using MEROPS [2]. All of the B. malayi aspartyl proteases identified by MEROPS were aligned with HIV-1 protease (Genbank accession number: CAA09224) using T-Coffee [30] and clustered with Clustal Omega [31]. The orthology was determined among the 6 nematode species with human using the Orthofinder with default settings [32]. Orthologous protein families annotated as aspartyl proteases in Brugia spp., Onchocerca spp. and T. muris were selected for multiple sequence alignment and construction of a phylogentic tree using the neighbor joining method was performed with MAFFT version 7 with default setting [33]. The alignment was visualized with Jalview [34].

Best bi-directional hits were identified based on protein alignment between the longest isoforms per gene of B. malayi and those of B. pahangi using Diamond blastp (v2.0.6.144) [35]. This alignment was performed in ‘both directions’ (once with B. malayi as the query and once with B. pahangi as the query) and a local script was used to build a table of results with all cases in which proteins were top scoring hits to each other in both direction and were flagged as reciprocal hits. Gene symbols and descriptions were harvested from the BioMart [36] instance hosted at WormBase Parasite [28] and were used to annotate the results

  1. Line 235: The supplementary table title is not correct. It is presented as "Supplementary Table S1: RNA-seq sample metadata including read mapping statistics and raw data accession identifiers." while the results are about % of inhibition of motility. Moreover, it would be important to explain what exactly the dash ("-") on the table is about. For instance, it is corresponding to "not tested" or "not active"?

Authors’ response: This has been corrected as follows:

We have also replaced BioSample numbers in 2nd column of Supplementary Table S2 with SRA accession number since GenBank recently assigned final accession IDs to our submission. 

  1. Figure 1A. It would be interesting to investigate the ritonavir different behaviour since it seems not to affect the worm in the first 3 days, being effective only after day 4. Could it be related to a previous metabolism by the nematode organism or, any necessary transformation? THIS POINT IS A SUGGESTION FOR FUTURE STUDIES, NOT REQUIRED TO APPROVE THIS PAPER.

Authors’ response: Thank you for this comment.  There may be redundant aspartyl proteases that are active in the worm, and it may several days for the ritonavir to exert its physiological effect.  The following text was added: Future studies will be needed to determine if the slower effect of ritonavir is in part due to the activity of other redundant aspartyl proteases.

  1. Finally, the authors suggest that the tested APIs would be suitable for drug repurposing, but I would be a little bit more careful since drug repurposing requires more than being active against a new target... It is also about being efficient and safe at dosages near those ones already used in the current therapy. For instance, in this paper, the authors are exploring anti-HIV drugs with known side effects with an impact on the patient's life quality. This same impact is acceptable for LF patients?

Authors’ response: We agree with this comment and recognize that additional studies are necessary to determine the therapeutic indices and physiological pharmacokinetics for these drugs as anti-filarial drugs. The last sentence in the CONCLUSION section now reads:

“Future in vitro studies are necessary to validate the target of these APIs, to optimize the API structures to increase potency, and to follow up with in vivo studies for safety and pharmacokinetics, eventually leading to in vivo studies in animal model for filarial and whipworm infection. Furthermore, additional studies are necessary to determine the therapeutic indices and physiological pharmacokinetics for these drugs as anti-filarial drugs. Overall, the results of this study suggest APIs may serve as new leads for drug discovery of filarial, gastro-intestinal and potentially other diseases.

Reviewer 2 Report

The article is well scientifically designed and well written.

There are some points that need to be addressed by the authors.

In Materials and Methods

The authors should explain why they used only female B. malayi worms.

Why 1% DMSO was the negative control?

How many well/ B. malayi worms per each replicate (how many replicates) were tested for each drug/concentration?

Mention the source and form of each APIs, also the used solution/media?

Why, during “Histochemical Localization of the Aspartyl Proteases in Adult Female B. malayi Immunohistologythe author used adult B. malayi recovered from experimentally infected gerbils, while “for In vitro Worm Motility Assays the authors used Adult female B. malayi from the FR3 at the University of Georgia.

Why did the authors not assess the death of the worms rather than motility, which may be reversed and parasite recover after removing the drug?

Results

In figure 3 specially3F “Wolbachiain the lateral chord, need to be magnified to recognise its morphology and document the absence of gold particles inside them.

Line 305: remove the word “are”

Author Response

The article is well scientifically designed and well written.

There are some points that need to be addressed by the authors.

In Materials and Methods

  1. The authors should explain why they used only female  malayiworms.

Authors’ response: Female Brugia spp were used in the in vitro assays to assess the potency of the compounds since adult female worms are the ones that release several thousand microfilariae over their lifetime and are responsible for maintaining the transmission cycle and spread of the disease.  Our aim is to identify potent drugs that eliminate the female worms and/or significantly reduce their fecundity.

  1. Why 1% DMSO was the negative control?

Authors’ response: The drugs are dissolved in 1% DMSO, therefore 1% DMSO was used as a control.

  1. How many well/ malayiworms per each replicate (how many replicates) were tested for each drug/concentration?

Authors’ response: Every well had a single female B. malayi worm and testing was done with 4 biological replicates.

  1. Mention the source and form of each APIs, also the used solution/media?

Authors’ response: The following text has been added to the Methods section:

10 mg of lopinavir (catalog #: S1380), nelfinavir (catalog #: S4282), and ritonavir (catalog #: S1185)  were purchased from Selleckchem. 10 mM stock solution were prepared with dimethyl sulfoxide (DMSO, Sigma-Aldrich D2650) and were diluted at different test concentrations in the culture media containing 1% DMSO as a final concentration.

  1. Why, during “Histochemical Localization of the Aspartyl Proteases in Adult Female B. malayi Immunohistology” the author usedadult  malayi recovered from experimentally infected gerbils, while “for In vitro Worm Motility Assays” the authors used Adult female B. malayi from the FR3 at the University of Georgia.

Authors’ response:  Washington University School of Medicine maintains its own B. malayi life cycle because only a limited number of worms can be ordered from FR3 per year. However, the FR3 strain of B. malayi and the parasites at WUSM are very closely related as they have the same ancestors and WUSM is sometimes obtaining blood that contains microfilariae from FR3 to increase L3 production in mosquitoes.

  1. Why did the authors not assess the death of the worms rather than motility, which may be reversed and parasite recover after removing the drug?

Authors’ response: Our in vitro assays have shown that inhibition of motility is highly correlated with worm morbidity and worm death based on the MTT ( 3-(4,5-dimethylthiazol-2-yl)-2,5-diphenyltetrazolium bromide) assay which is used to measure cellular metabolic activity and is an indicator of cell viability. After worms have stopped moving, they are also assessed for their morphology as well, e.g. internal structures appear opaque and shrunken.  We have also conducted “washout” experiments in vitro to demonstrate that once the worms stop moving after a day, they do not recover.  There are exceptions, however, such as the drug levamisole, which paralyzes the worms and then they recover.

Results

  1. In figure 3 specially3F “Wolbachia” in the lateral chord, need to be magnified to ecognize its morphology and document the absence of gold particles inside them.

Authors’ response: We have split up the panels in Figure 3 (C-H) as a separate morphology multi-panel figure which includes magnified 3F panel (now panel D).

  1. Line 305: remove the word “are”

Authors’ response: The word ‘are’ has been removed.
